# Tribological Properties and Corrosion Resistance of Stellite 20 Alloy Coating Prepared by HVOF and HVAF

**Zhiqiang Zhou [1,2,*], Jiahui Yong [1,2], Jiaoshan Hao [1,2], Deen Sun [3], Qian Cheng [2], Huan Jing [2] and Zhongyun Zhou [2]**

1  Technology Center Regulating Valve Research Institute, Chongqing Chuanyi Automation Co., Ltd., Chongqing 400707, China
2  Chongqing Chuanyi Control Valve Co., Ltd., Chongqing 400707, China
3  School of Materials and Energy, Southwest University, Chongqing 400715, China
*  Correspondence: zhouzzq6@gmail.com

**Abstract:** This study examines the tribological and corrosion properties of Stellite 20 alloy coatings on F310H heat-resistant stainless steel that were prepared using HVOF and HVAF supersonic flame spraying techniques. To investigate the coatings' microstructure, phase, microhardness, wear, and corrosion resistance, a range of characterization techniques, including SEM, EDS, XRD, microhardness, and friction wear-testers, weas employed. The results indicate that both HVOF and HVAF-prepared coatings exhibit a dense structure with porosity of 0.41% and 0.32%, respectively. The coatings are composed of $\gamma$-Co solid solution, $\varepsilon$-Co solid solution, Cr-rich solid solution, $Cr_7C_3$, WC, and $CoCr_2O_4$ phases. The microhardness of the Stellite 20 coatings prepared by HVOF and HVAF methods was 610 $HV_{0.3}$ and 690 $HV_{0.3}$, respectively, which is three times higher than that of the F310H stainless steel substrate. The wear mechanism of the HVAF coating is abrasive wear, while the wear mechanism of the HVOF coating is mainly fatigue wear with slight abrasive wear. The HVAF coating demonstrates superior wear resistance due to its higher flame velocity, denser coating, and higher average microhardness. In contrast, the HVOF coating shows a higher friction coefficient stability due to its lower hardness dispersion. The corrosion potentials of the HVOF and HVAF coatings are $-0.532$ V and $-0.376$ V, respectively, with corresponding corrosion current densities of $1.692 \times 10^{-7}$ A·cm$^{-2}$ and $6.268 \times 10^{-7}$ A·cm$^{-2}$, respectively. Compared to the HVOF coating, the Stellite 20 coating prepared using HVAF technology exhibits better wear and corrosion resistance.

**Keywords:** supersonic flame spraying; HVAF; HVOF; wear mechanism; corrosion resistance mechanism

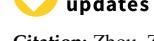



## 1. Introduction

With the increasing demands for environmental protection, waste incineration power generation has become the mainstream waste treatment technology. The incineration of garbage produces harmful and toxic gases at temperatures ranging from 800 to 1000 °C. However, by using dust removal equipment to purify the gases, the resulting heat can be recovered through a waste heat boiler to produce high-temperature and high-pressure steam. This steam can then be used to generate electricity, achieving the reutilization of waste resources [1]. The flue gas from the waste incineration furnace contains numerous corrosive substances, such as sulfur dioxide, chlorides, and sulfates [2–4], as well as metal dust and solid particles. Valves, as important gas flow control devices on the pipeline of this equipment, are often subjected to erosion by high-temperature and high-pressure gases. In order to improve the service life of the valves, surface hardening technology has become a critical research topic in addressing the wear and corrosion resistance of valve cores and valve seats under high-temperature and high-pressure conditions. These components must resist the corrosive effects of substances such as sulfur dioxide, chlorides, and sulfates in flue gas purification equipment. Additionally, the presence of metal dust and solid particles can cause significant wear, leading to the failure of key components

and, ultimately, the entire system. Therefore, studying heat-resistant coating materials and thermal spraying technology is crucial for ensuring the long-term reliability and efficiency of flue gas purification equipment in waste incineration power plants.

Stellite alloy is a commonly used cobalt-based superalloy, which is composed of a brittle, hard phase and a tough bonding matrix phase [5]. Stellite cobalt-based alloys contain 20–30 wt.% Cr element, 4–18 wt.% W or Mo element, and 0.25–3 wt.% C element. The carbide is deposited in the bonding phase as a hard phase, ensuring that Stellite alloys have excellent wear resistance, hardness, and corrosion resistance at high temperatures. In addition, the martensitic phase transformation ($\gamma \rightarrow \varepsilon$) occurs during the coating preparation process [6,7], which makes the lower stacking fault energy of the cobalt-based alloy. At the same time, the residual $\gamma$ phase hinders dislocation movement [6], further improving the wear resistance of the coating. Therefore, Stellite alloy is widely used in metallurgy, petroleum, chemical, aerospace, and other fields [8].

Common Stellite alloy hardening processes include plasma welding [9,10], laser cladding [11,12], and thermal spraying technology [13–15]. Stellite alloy coatings prepared by different hardening processes have different microstructures and wear resistance properties [8,16]. Compared with the original alloy, the composition of Stellite alloy coatings prepared by plasma welding or laser cladding process will change due to dilution and dissolution of the matrix material, resulting in a decrease in the phase transformation ability and wear resistance of the coating [17].

High-Velocity Oxygen Fuel Spraying (HVOF) technology is widely used for surface hardening. Compared with other surface hardening processes, HVOF has the advantages of extremely low heat input, high flame velocity, and relatively low temperature, making it an important technology for preparing high-performance wear-resistant and corrosion-resistant coatings [17,18]. Therefore, under a variety of complex industrial operating conditions, such as surface processing of wear-resistant and corrosion-resistant conditions, including pumps, valves, impellers, and bearings, HVOF technology is extensively used and highly valued by the academic community.

However, the HVOF technology uses oxygen as the combustion-assisting gas, and the metal powder particles are in a rich oxygen atmosphere during the spraying process, which is prone to thermal decomposition of powder oxidation or carbides. High-Velocity Air–Fuel Spraying (HVAF) technology is a new technology developed in recent years. HVAF uses compressed air instead of expensive oxygen as the combustion-assisting gas and adopts a gas cooling method. This not only greatly reduces costs but also controls the spraying temperature within a lower range. It has been reported that the spraying flame velocity and flame temperature of HVAF technology are 700–1200 m/s and 1800 °C, respectively [19], while the spraying flame velocity and flame temperature of HVOF technology are 500–800 m/s and 3000 °C, respectively [20]. Therefore, HVAF has a higher spraying flame velocity and lower flame temperature than HVOF technology, which helps to form metal coatings with high density, low oxide content, and high bonding strength [21,22].

At present, research on the performance of Stellite coatings, usually prepared by plasma welding and laser cladding technology, encompasses a range of alloys, including Stellite 6, Stellite 12, and Stellite 21 [13]. Stellite 20 alloy combines high hardness and excellent corrosion resistance due to its high carbide content (usually exceeding 1% [23]) and W content exceeding 15%. Limited research has been conducted on the preparation of Stellite 20 coatings using supersonic flame spraying technology. The Stellite 20 coating prepared using HVOF and HVAF technology in this paper maintains its excellent wear and corrosion resistance due to the low heat input during the preparation process. In this paper, the microstructure, hardness distribution, wear resistance, and corrosion resistance of Stellite 20 coating prepared by HVOF and HVAF technology on the surface of AISI F310H heat-resistant stainless steel are systematically studied, aiming to provide a solution and theoretical support for industrial applications under high temperature and corrosion conditions in metallurgy, chemical, power industry, and other fields in the future.

## 2. Materials and Methods

### 2.1. Coating Preparation

The substrate material utilized in this research was AISI F310H high-temperature resistant stainless steel, which underwent solution treatment, resulting in an average of 195 $HV_{0.3}$. The specimen size was $\Phi$24.5 mm × 5 mm, and the surface roughness was Ra 0.4. The chemical composition of the substrate is shown in Table 1, which was determined using inductively coupled plasma optical emission spectrometry (ICP-OES) and a carbon sulfur analyzer (CSA). The spraying powder used was Stellite 20 alloy powder obtained from Kennametal Shanghai Company. The composition of the spraying powder is indicated in Table 2.

**Table 1.** Chemical composition of F310H stainless steel (wt.%).

| C | Mn | Si | S | P | Ni | Cr | Fe |
|---|---|---|---|---|---|---|---|
| 0.09 | 1.86 | 0.63 | 0.02 | 0.03 | 21.17 | 25.53 | Bal. |

**Table 2.** Chemical composition of Stellite20 alloy powder (wt.%).

| C | Cr | Fe | Mn | Mo | Ni | P | S | Si | W | Co |
|---|---|---|---|---|---|---|---|---|---|---|
| 2.43 | 32.47 | 1.47 | 0.1 | 0.09 | 1.61 | 0.01 | 0.014 | 0.63 | 17.31 | Bal. |

The high-temperature stainless steel substrate material, AISI F310H, was subjected to a rigorous cleaning process prior to spraying. To remove any impurities, such as oxide or oil residues that may remain on the surface of the sample, the F310H substrate was repeatedly cleaned using acetone and absolute ethanol and dried with compressed air. After cleaning, the substrate was sandblasted to ensure optimal surface roughness. The substrate was coated with a layer of Stellite 20 alloy using both the PRAXAIR JP8000 K2 HVOF (Indianapolis, IN, USA) and Unique Coat M2 HVAF (Oilville, VA, USA) supersonic flame spraying systems, respectively. The specific parameters used during the spraying process are detailed in Table 3. The Schematic presentation of HVAF/HOVF spraying technology is shown in Figure 1. Oxygen is used as a combustion-assisting gas to heat micron-sized or nanometer-sized metal ceramic powder particles along the axial. The melted or partially melted powder particles are accelerated and impacted on the substrate surface, spreading and solidifying rapidly, and then layer by layer, forming a coating with high bonding strength, good density, and excellent wear and corrosion resistance.

In order to obtain a smooth coating metallographic section, wire cutting was used to extract samples at the cross-section of the coating. These samples were then subjected to hot mounting, rough grinding, fine grinding, and polishing to achieve a surface roughness of Ra < 0.1 μm.

**Table 3.** Process parameters of HVOF and HVAF.

| HVOF | Oxygen flow (L/min) | Kerosene flow (L/h) | Nitrogen flow (L/min) | Air flow (m$^3$/min) | Spraying distance (mm) | Powder speed (mm/s) | Powder feeding (g/min) |
|---|---|---|---|---|---|---|---|
| | 1820 | 21 | 20 | 10 | 370 | 720 | 110 |
| **HVAF** | Air pressure (MPa) | Propane pressure (MPa) | Nitrogen flow (L/min) | Air flow (m$^3$/min) | Spraying distance (mm) | Powder speed (mm/s) | Powder feeding (g/min) |
| | 0.54 | 0.49 | 60 | 20 | 230 | 800 | 110 |

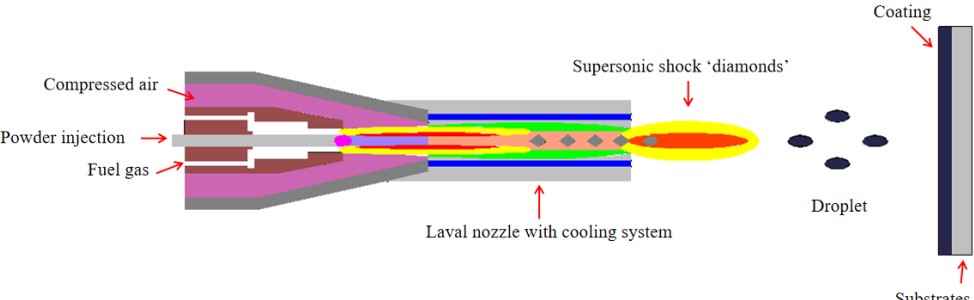

**Figure 1.** Schematic presentation of HVAF/HOVF spraying technology.

*2.2. Performance Characterization*

The Navo Nano SEM450 (FEI, Hillsboro, OR, USA) field emission scanning electron microscope (SEM) was utilized to observe the powder morphology, surface, cross-section, and coating morphology following friction and wear testing. The chemical composition of the coating was analyzed by Quantax-200 EDS (Bruker, Billerica, MA, USA) spectrometry, and the particle size distribution of the powder was determined using a laser particle size analyzer (Microtrac S3500, Largo, FL, USA). The coating porosity was calculated according to ASTM E-2109-01 standard [24] by measuring the area of voids in the coating cross-section using Image J image processing software. Five measurements were taken in different regions and averaged.

The microhardness of the coating was measured using the INNOVATEST FALCON 500 (Eindhoven, The Netherlands) Vickers hardness tester. ASTM E-384-89 [25] with a loading load of 300 g and a loading time of 15 s were used to measure 15 points in the field of view area of the coating cross-section, according to the normal distribution curve. The hardness distribution characteristics of the coating were investigated using Weibull distribution, and Equations (1) and (2) were employed to characterize the hardness distribution characteristics of the coating [26].

$$\ln\{-\ln[1 - \mathrm{F}(x)]\} = \beta \ln x + \ln \varphi \tag{1}$$

$$\mathrm{F}(x) = \frac{i - 0.5}{n} \tag{2}$$

A linear expression is obtained by fitting a regression line to the discrete data using linear regression, which involves plotting $\ln x \sim \ln\{-\ln[1 - \mathrm{F}(x)]\}$ coordinate points and determining the equation of the line that best fits the data:

$$y = kx + b \tag{3}$$

where $x$ is the hardness value and $\beta$ and $\ln \varphi$ are the values of the parameters and are the slope and y-intercept of the line on the axis, respectively. $i$ is the index corresponding to the hardness values sorted in ascending order, $n$ is the number of experiments, and $\beta$ is the modulus of the Weibull distribution.

The Rigaku D/MAX 2500 PC (Tokyo, Japan) X-ray diffractometer (XRD) was used to analyze the phase of the powder and coating by selecting Cu target K$\alpha$ radiation with a voltage of 40 kV, current of 300 mA, and a scan speed and step size of 6 ($^\circ$/min) and 0.02 ($^\circ$), respectively.

The friction and wear test of the coating was carried out according to ASTM G99-05 [27] standard. Prior to testing, the sample surface was sequentially polished with 220, 500, 800, and 1200 grit SiC sandpaper and $Al_2O_3$ suspension to achieve a surface roughness (Ra) below 0.1 μm. The MFT-5000 (Rtec Instruments, San Jose, CA, USA) multifunctional friction and wear tester was used to test the friction and wear performance of the coating. In this study, a 9.5 mm diameter WC ceramic ball was used as the friction pair, and the load, frequency, stroke, test environment temperature, humidity, and test time were set to

5 N, 2 Hz, 5 mm, 28 °C, 40%, and 10 min, respectively. Three samples were performed for each coating.

Electrochemical measurements were conducted in a three-electrode cell using a platinum counter electrode, the sample as the working electrode and a saturated calomel reference electrode (SCE). The samples for the corrosion test were sealed with epoxy resin around, leaving only an end surface (with a surface area of about 1 cm$^2$) exposed for testing. The potentiodynamic polarization curves were recorded at a sweep rate of 2 mV/s in 3.5% NaCl solution open to air at room temperature. During the measurement of the potentiodynamic polarization curve, three repeated experiments were conducted for each coating.

## 3. Results and Discussion

### 3.1. Powder Morphology

The microstructure of the Stellite 20 alloy powder raw material prepared by gas atomization is shown in Figure 2a. The high sphericity of the powder indicates good flowability of the powder. Figure 2b shows that the particle size distribution of the powder raw material conforms to the Gaussian curve, with $d_{mean}$ = 35 μm, $d_{10}$ = 20 μm, and $d_{90}$ = 53 μm. The different particle size distributions of the powder can have a significant impact on the various physical properties of thermal spray coatings [28,29]. Therefore, it is important to select powder raw materials with appropriate particle size distributions when using different thermal spray processes and process parameters [30]. To ensure reliable and stable comparability of coating performance for comparative studies, this study used raw powder materials with the same particle size distribution for coating preparation while using the most suitable HVOF and HVAF thermal spray process parameters.

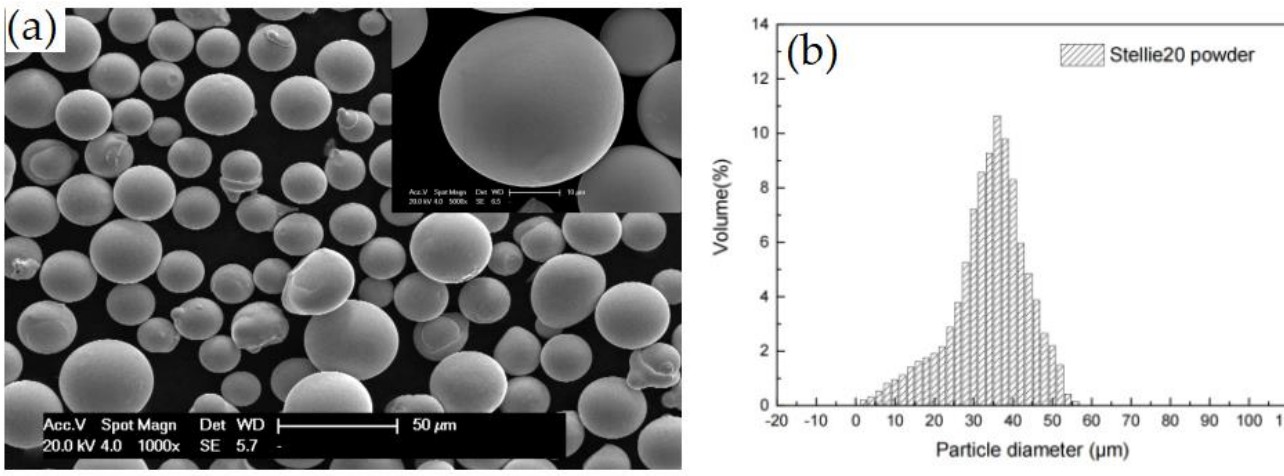

**Figure 2.** (**a**) The morphology of Stellite 20 powder and (**b**) particle size distribution of the feedstock powder.

### 3.2. Surface Morphology

The surface morphology of Stellite 20 coatings prepared by HVAF and HVOF is shown in Figure 3. The surface morphology of HVAF coatings, as shown in Figure 3a,c, exhibits a large number of fragmented particles and partially melted zones, as well as a small amount of fully melted zones, with a high surface roughness. This behavior can be attributed to the HVAF flame temperature of approximately 1800 °C, which is slightly higher than the melting point of the cobalt bonding phase in Stellite 20 alloy. Moreover, the HVAF flame velocity is approximately 1.5 times higher than that of HVOF, causing the powder to undergo strong plastic deformation upon impact on the substrate surface. This process impedes the formation of molten or semi-molten droplets, and the powder is then stacked layer by layer to form a coating.

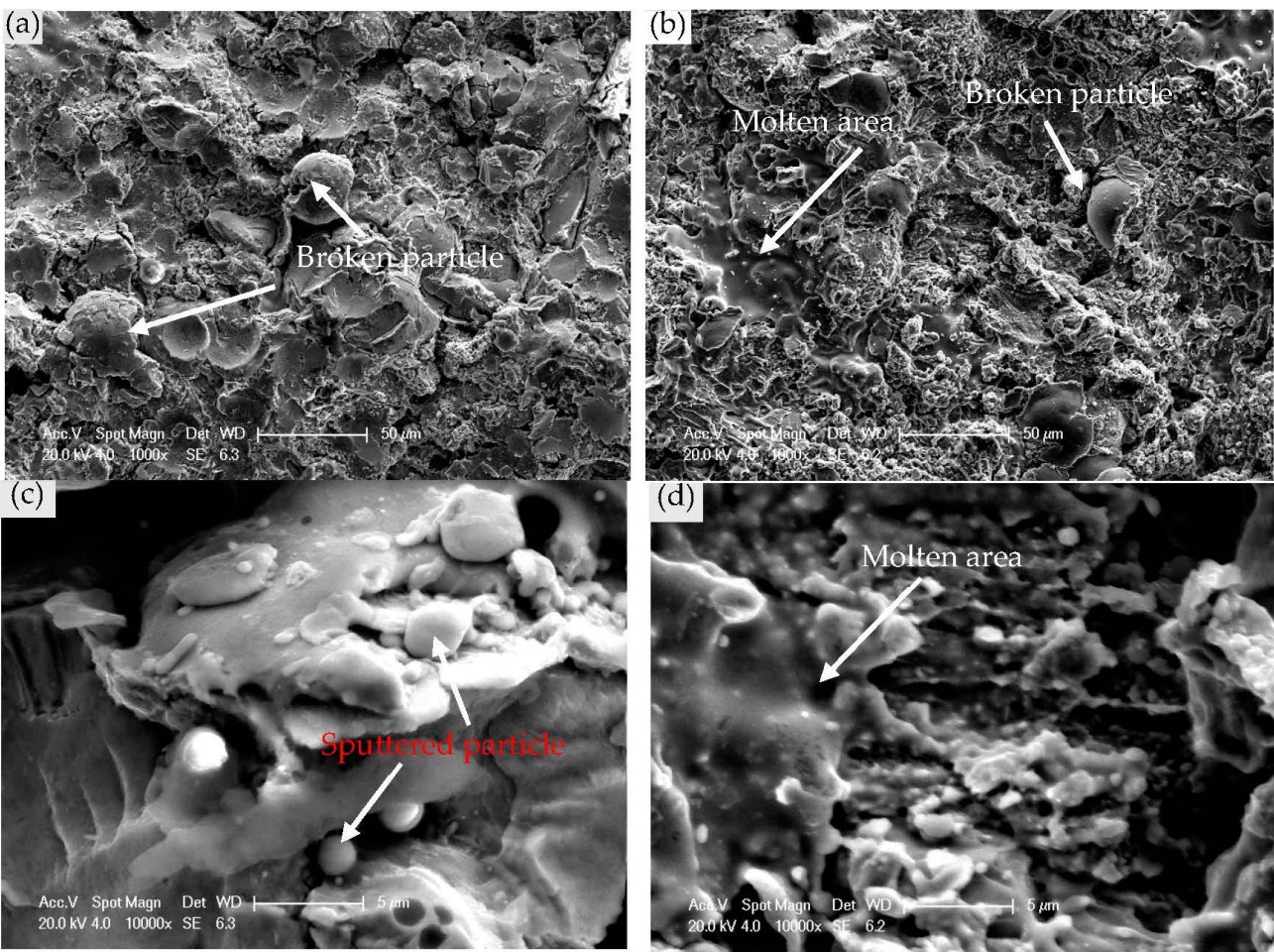

**Figure 3.** (**a**,**c**) are the morphology and high magnification picture of the as-sprayed Stellite 20 coating prepared by HVAF; (**b**,**d**) are the morphology and high magnification SEM image of the as-sprayed Stellite 20 coating prepared by HVOF.

On the other hand, Figure 3b,d illustrate that the HVOF coating exhibits a uniform and relatively smooth surface structure, consisting of numerous partially melted and fully melted regions, with only a small quantity of fragmented particles. In contrast to ceramic spraying powders, such as WC and $Cr_7C_3$, which possess higher melting points, Stellite 20 alloy, primarily composed of cobalt metal powder, has a relatively low melting point of approximately 1500 °C. As a result, in the HVOF spraying process, most of the metal powder is heated above its melting temperature, as the flame temperature of HVOF is approximately 3000 °C. The molten droplets hit the substrate surface at an extremely high velocity, solidify and stack layer by layer, ultimately generating a dense and bonded coating.

*3.3. Section Morphology*

Figure 4 shows the SEM cross-sectional morphology of Stellite 20 coatings prepared by HVAF and HVOF. The experimental results indicate that the thickness of Stellite 20 coatings prepared by HVAF and HVOF is 0.25–0.30 mm. Additionally, the coatings exhibit a highly compact structure with a typical thermal spray layer formation. This formation is due to the flattening and spreading of high-velocity molten particles that collide with the substrate and subsequently cool down. In Figure 4, numerous black contours can be observed at the junction of completely molten particles. The EDS spectrum analysis of these contours (Table 4) reveals the presence of O elements, indicating the occurrence of a minor oxide film between the layered structures. This is because some of the molten metal powder particles in the flame flow come into contact with the surrounding air and undergo oxidation,

forming a layer of oxide film enveloping the particle surface. However, the high velocity of the particles in the flame flow limits the contact time with the air, resulting in minimal oxidation and thin oxide film thickness.

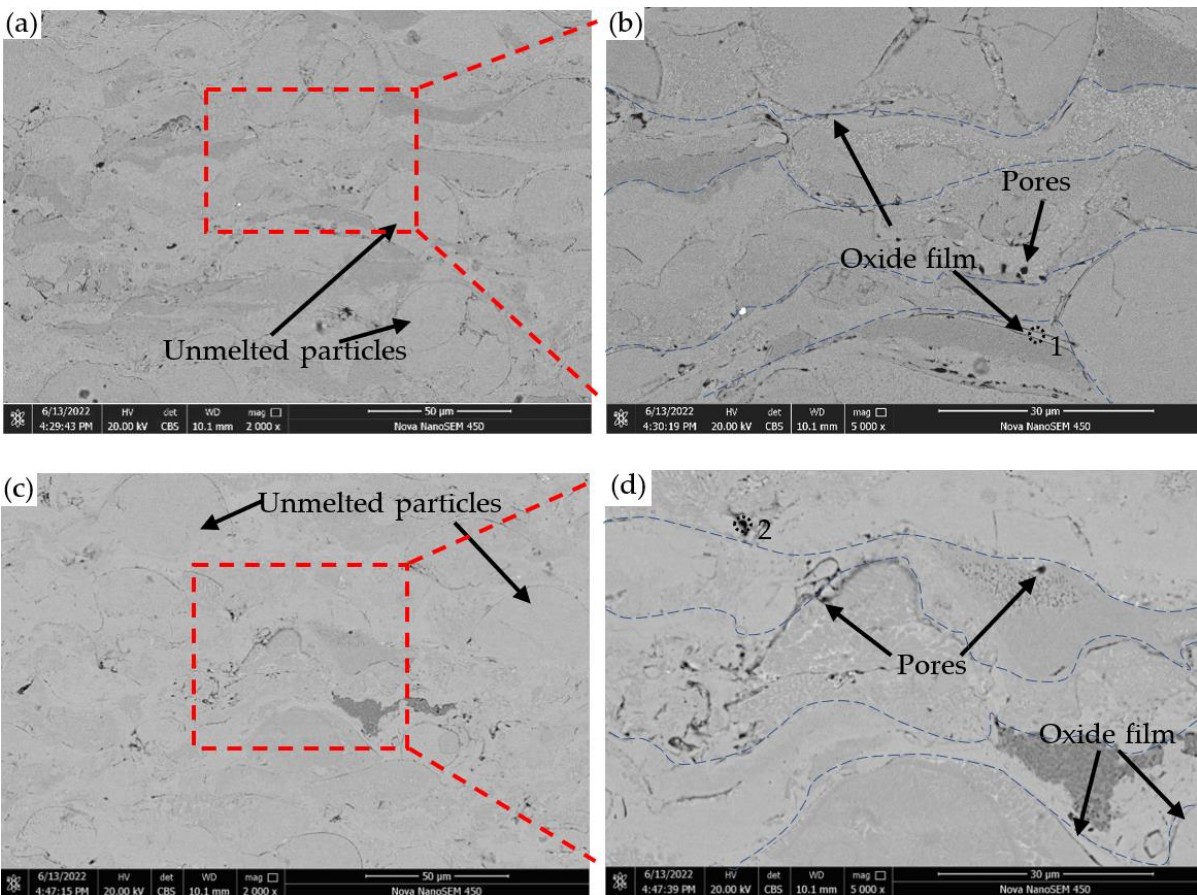

**Figure 4.** (**a,b**) Section morphology of Stellite 20 coating deposited by HVAF; (**c,d**)Section morphology of Stellite 20 coating deposited by HVOF.

**Table 4.** EDS results of the corresponding areas of section morphology (at%).

| Areas | Element | | | | | | | | | | |
|---|---|---|---|---|---|---|---|---|---|---|---|
| | O | Co | Cr | C | Fe | W | Ni | Mo | Si | Sr | Ti |
| 1 | 3.63 | 23.62 | 23.41 | 38.62 | 1.62 | 3.40 | 1.67 | 1.09 | 1.75 | 1.19 | - |
| 2 | 3.98 | 22.24 | 18.29 | 49.85 | 1.43 | 2.58 | 1.40 | - | - | - | 0.23 |

By observing the cross-sections of the HVAF coatings in Figure 4a,b, it is apparent that there are interfacial boundaries and small pores, as well as a large number of unmelted particles that are broken up and enclosed by the melted Co-based binder phase with a low melting point. This phenomenon is a result of the HVAF process's lower flame temperature during spraying, which causes some metal powder particles to fail to melt completely before impacting the substrate surface at high velocities.

In contrast, Figure 4c,d illustrate the cross-sectional morphology of HVOF coatings, which have a dense structure and exhibit fewer unmelted particles in comparison to HVAF coatings, with particles being more fully melted and flattened. This can be attributed to the fact that in the HVOF process, most of the powder is deposited in the form of liquid droplets on the substrate or bottom layer surface and solidifies into a coating. Additionally, the particle flight speed is relatively low, resulting in less breakage and more uniform distribution within the Co-based binder phase.

Figure 4 demonstrates that the coating pores are primarily found at the interface of fragmented particles, and some pores are distributed around the bonding phase due to incomplete compensation for the solidification shrinkage of the fully melted bonding phase. The "gray-scale method" was used to accurately assess the effect of HVAF and HVOF spraying processes on the coating porosity of Stellite 20 coatings, resulting in average values of 0.32% and 0.41%, respectively, indicating that both methods produce coatings with high density.

Figure 5 exhibits the X-ray diffraction (XRD) spectra of Stellite 20 coatings prepared by HVOF and HVAF, respectively, as well as the initial powder. The Stellite 20 coating is primarily composed of face-centered cubic (fcc) cobalt-based solid solution (i.e., $\gamma$-Co solid solution) and a small quantity of densely packed hexagonal (hcp) filling phase (i.e., $\varepsilon$-Co solid solution) formed during spraying. Meanwhile, a Cr-rich solid solution was also obviously observed. The presence of two distinguishable solid solutions in the coatings can be ascribed to particles' melting within the flame and to the extremely high cooling rates subsequently experienced by the fully melted particles when impacting the cold substrate. A low-intensity peak is observed in the XRD spectrum, which can be identified as $M_7C_3$ carbide (WC, $Cr_7C_3$). The presence of this carbide is in agreement with the Co-Cr-C phase diagram, which predicts the equilibrium between $M_7C_3$ and $\gamma$-Co solid solution, i.e., the theoretical composition of Co-28%Cr-1.1%C (wt.%) [31]. Only small amounts of $M_7C_3$ are visible in the powder and coating XRD spectra owing to the high cooling rate of the powder during gas atomization and spraying, which reaches $10^5$ °C/s [32], inhibiting the precipitation of the $M_7C_3$ carbide second phase. During supersonic spraying, the Co (hcp) content in the Stellite 20 coating marginally increases relative to the initial powder, implying the occurrence of martensitic phase transformation ($\gamma \rightarrow \varepsilon$) during the coating preparation process [7,8]. These outcomes suggest that the microstructure of the Stellite 20 coating is influenced by the spraying process, and its primary composition phase is identical to that of the initial powder but with slight variations.

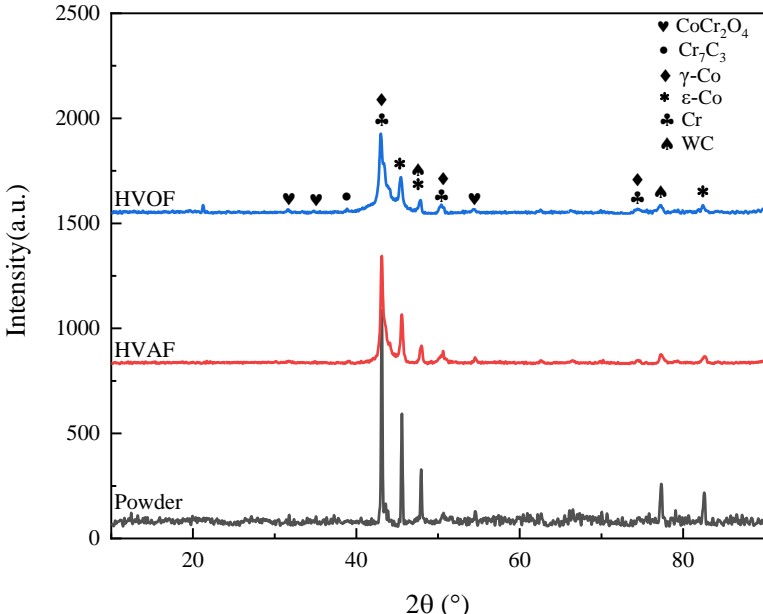

**Figure 5.** XRD patterns of HVAF and HVOF sprayed coatings.

Through a comparison of the X-ray diffraction (XRD) patterns of the raw powders and the HVOF/HVAF coatings, it was observed that all diffraction peaks in the HVOF/HVAF coatings exhibited significant broadening [13,33]. Extant research indicates that this broadening phenomenon arises from the micro-stresses engendered during the powder deposition process [34] and the grain refinement that occurs during the high-speed cooling of the melted powders. Furthermore, from the XRD patterns, a small quantity of $CoCr_2O_4$

oxide was detected in the coatings, suggesting that the metal powders underwent slight oxidation during the supersonic spraying process.

### 3.4. Microhardness and Weibull Distribution

Figure 6 displays the microhardness distribution of Stellite 20 coatings that were prepared by HVAF and HVOF methods. The microhardness of the Stellite 20 coating was found to be three times higher than that of the F310H substrate. The microhardness near the interface between the coating and substrate increased slightly to about 350 $HV_{0.3}$, indicating that the powerful impact during powder deposition had a certain work-hardening effect on the substrate. The microhardness of the HVAF coatings ranged from 600 to 800 $HV_{0.3}$, with an average value of 690 $HV_{0.3}$, whereas that of the HVOF coatings ranged from 600 to 700 $HV_{0.3}$, with an average value of 601 $HV_{0.3}$. As noted in a previous study [35], during supersonic spraying, Stellite powder particles underwent strong plastic deformation, leading to the formation of a large number of twin crystals in the coating that increased its microhardness. However, due to the high flame temperature of the HVOF technology and the more complete melting of the powder, the hard phase broke and dissolved into the binder matrix, thereby reducing the microhardness of the coating. Hence, the average microhardness of the HVAF coatings was higher than that of the HVOF coatings.

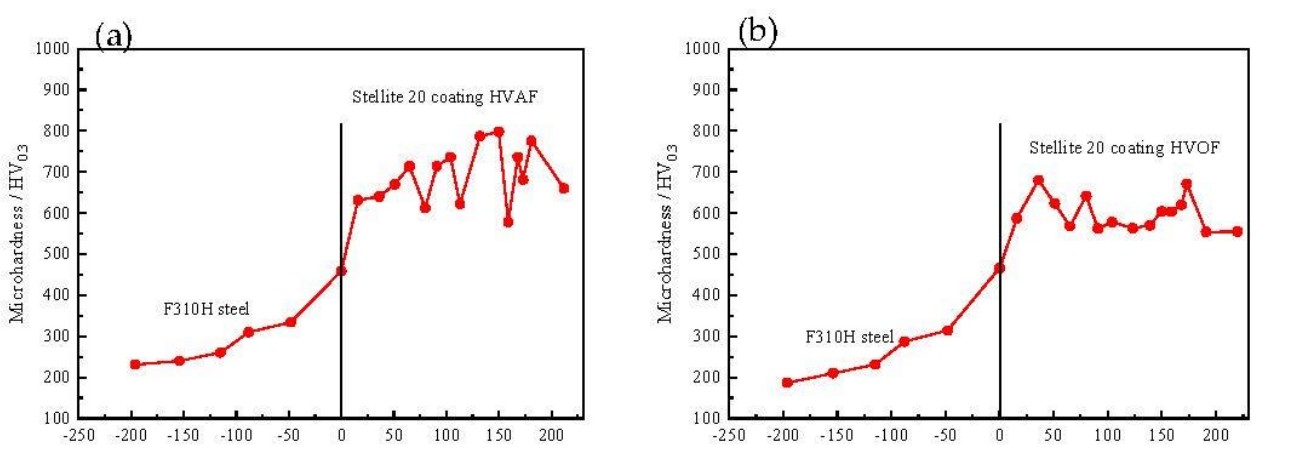

**Figure 6.** Microhardness profile of Stellite 20 coatings deposited by HVAF (**a**) and HVOF (**b**).

In order to address the issue of large dispersion in determined microhardness values, which can be attributed to the non-uniformity of composition and microstructure of thermal spray coatings, as well as errors in sample preparation and measurement, this study employs the Weibull statistical method [25] to further characterize the relationship between the microstructure and microhardness of supersonic sprayed coatings. The shape parameter (β) in equation (1) reflects the dispersion of coating hardness distribution, with a larger value indicating lower dispersion and higher stability of coating properties, and vice versa.

The Weibull distribution curves of microhardness for HVAF and HVOF coatings are shown in Figure 7. Both coatings exhibit a single peak distribution feature under a 300 g load, which means that there is only one peak point on the distribution curve, indicating that the hardness of the coatings has a relatively concentrated range of values and follows the assumption of Weibull distribution. This performance shows that the microhardness of the coatings is relatively stable within a certain range. The β of Stellite 20 coating prepared by HVAF is 9.27, with a wide distribution range of microhardness values and a large range of extreme values, indicating relatively lower mechanical stability of the coating. In contrast, the β of HVOF coating is 13.96, with a narrow distribution range of microhardness values and a small range of extreme values, indicating a more stable mechanical performance of the coating. This can be attributed to the higher flame temperature of the HVOF process, which enables sufficient melting of powder particles. Partially melted or unmelted particles

are dissolved in the fully melted matrix or low melting point Co-based binder phase during the spraying process, resulting in a more uniform microstructure distribution and higher mechanical stability of the coating. In contrast, the lower flame temperature of the HVAF process leads to a relatively higher number of partially melted or fractured particles, which are randomly distributed in the fully melted matrix, resulting in the lower mechanical stability of the coating.

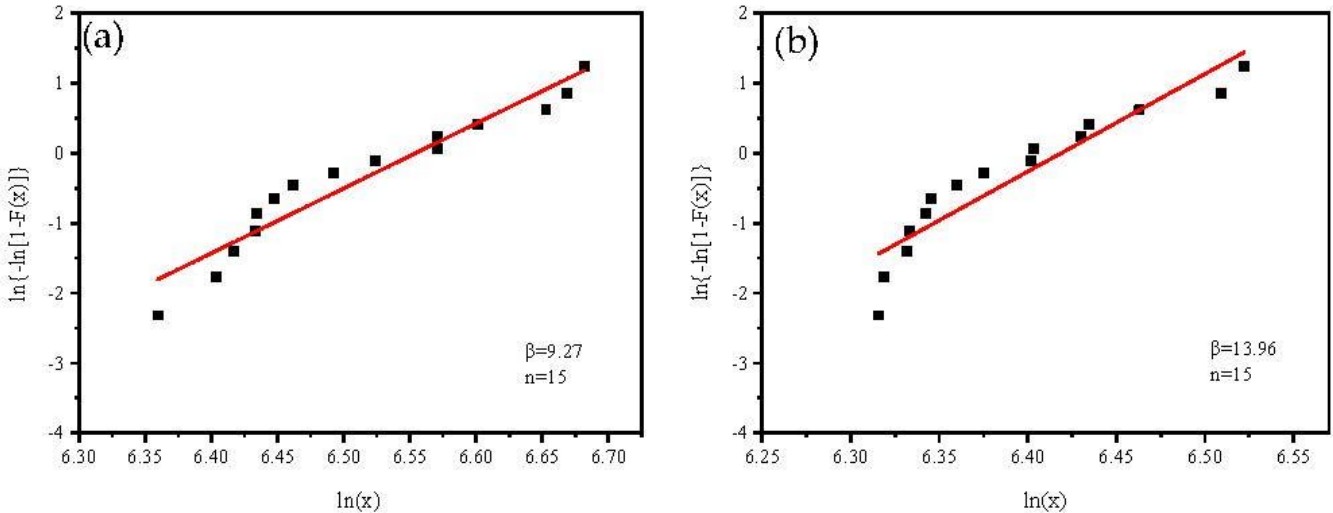

**Figure 7.** Weibull plots of mirohardness for Stellite 20 coatings deposited by HVAF (**a**) and HVOF (**b**).

### 3.5. Wear Behavior

#### 3.5.1. Friction Coefficient and Wear Volume

Figure 8 illustrates the friction coefficient curves of Stellite 20 coatings prepared by HVAF and HVOF techniques under dry sliding conditions. As depicted in the figure, the friction coefficients of the coatings display an initial running-in stage followed by a stable period. During the running-in stage, the mating material shears the micro-convexities present on the coating surface. Due to the loose structure of the micro-convexities, the contact area between the mating material and the coating surface rapidly increases, leading to a sharp surge in the friction coefficient of the coating. As the friction and wear test progresses, the micro-convexities are worn down, and the coating surface becomes comparatively smoother. Nonetheless, the presence of hard phases and defects, such as pores, in the coating results in fluctuations in the friction coefficient within a certain range during the stable period.

The friction coefficient of the HVAF coating initially increases rapidly and then drops sharply within the duration of 0–90 s. Afterward, the coefficient increases slowly and tends to be stable with minor fluctuations, with an average friction coefficient of 0.4483. The friction coefficient of the HVOF coating increases rapidly during the running-in period and gradually decreases afterward, eventually tending to be stable with an average friction coefficient of 0.3974. During the stable period, the friction coefficient of the HVOF coating exhibits higher stability compared to the HVAF coating, which can be attributed to the lower dispersion of hardness in the HVOF coating. This, in turn, enables the HVOF coating to display exceptional stability.

The 3D morphology of the wear tracks and wear volume of Stellite 20 coatings prepared by HVAF and HVOF are presented in Figure 9. The observed wear tracks of Stellite 20 coatings show typical groove shapes, which are attributed to the groove-shaped wear generated by the tungsten carbide (WC) in the process of shear and cutting of the microsurface between the mating pair and the coating [18,19]. Furthermore, conspicuously raised accumulations on both sides of the wear marks are observed, indicating that the alloy material does not directly fracture during the friction process but is continuously extruded along both sides, which reflects the good toughness of the alloy [25].

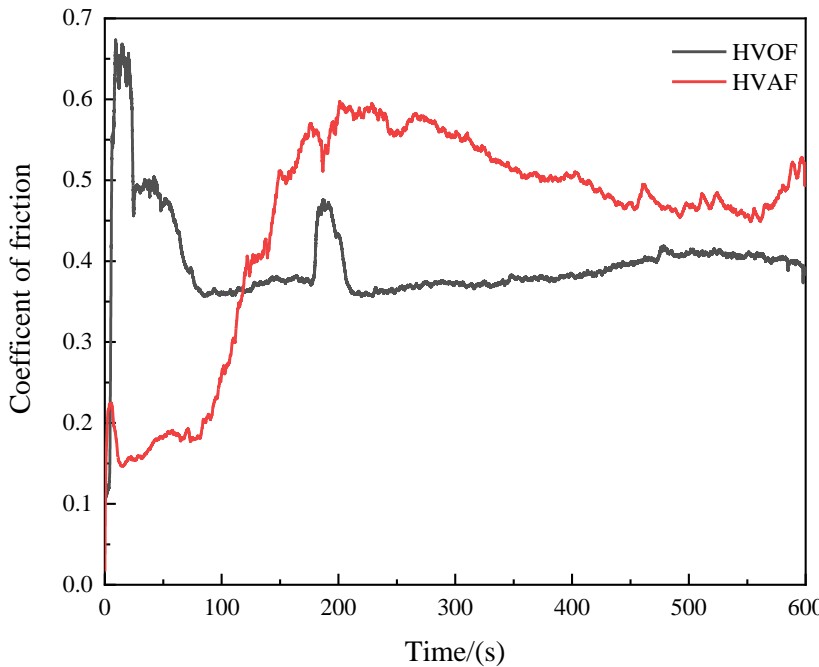

**Figure 8.** Friction coefficient curve of Stellite 20 coating deposited by HVAF and HVOF.

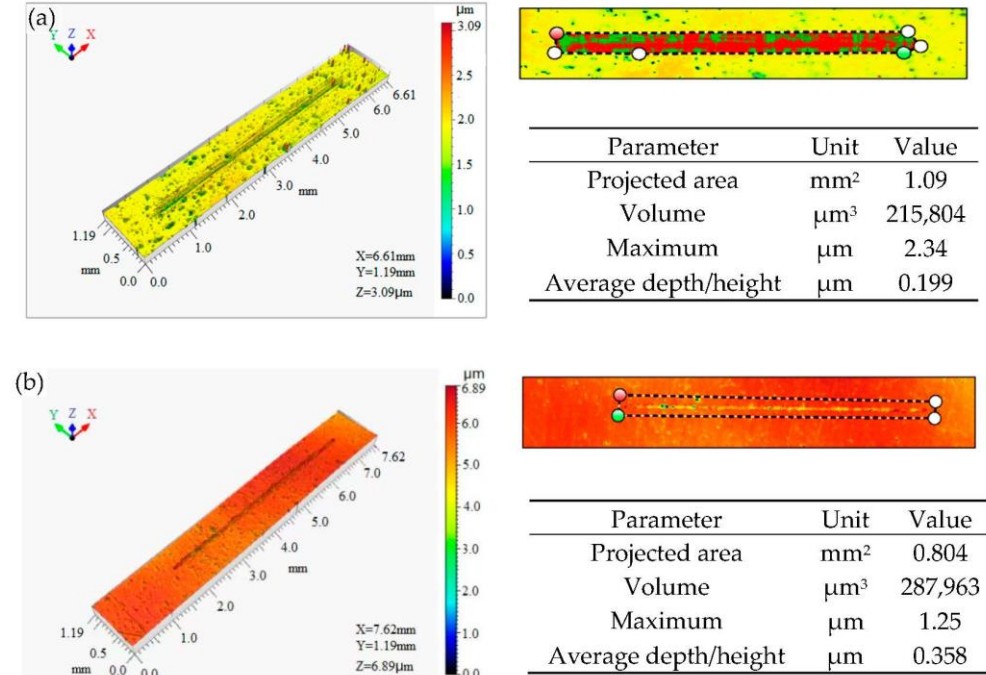

**Figure 9.** Three-dimensional wear track morphologies and wear volume of Stellite 20 coating: (**a**) HVAF spray coating; (**b**) HVOF spray coating.

The average depth of the wear marks on the HVAF coating shown in Figure 9 is 0.199, which is lower than the average depth of wear marks on the HVOF coating (0.358). Moreover, the cumulative wear volume of the HVAF coating is 25% lower than that of the HVOF coating. As indicated by the results in Figures 4 and 5, this is due to the HVAF coating undergoing a certain degree of grain refinement and containing more hard-phase particles and fewer oxides, which enables it to better resist wear from the mating surface. Thus, the HVAF coating exhibits better wear resistance, with smaller wear marks and volumes.

It is worth noting that the width of the wear marks on the HVAF coating in Figure 9a is wider than that of the HVOF coating. This could be attributed to the HVAF coating having a relatively higher hardness, causing the WC counterpart to undergo more wear during the sliding process than it does on the HVOF coating, resulting in a wider wear mark on the HVAF coating.

### 3.5.2. Wear Mechanism

Figure 10 shows the wear morphology of Stellite 20 coatings prepared by HVAF and HVOF under a 5 N load. The coatings exhibit evident plastic deformation zones on their surfaces, with continuous and complete scratches. The surface of the HVAF coating, shown in Figure 10a,b, is characterized by distinct furrows, indicating abrasive wear. Additionally, pits are present on the surface of the coating due to the layered structure of supersonic flame spraying coatings, formed by numerous deformed particles interlocking, overlapping, and stacking. During the friction and wear process, the wear initially originates from the lower hardness Co bonding phase. Under the action of normal loading force, WC is pressed into the coating, causing hard particles to protrude from the surface of the coating. Subsequently, as the hard particles move relative to the counter surface, a considerable tangential friction force is generated, leading to the cutting of the micro-protrusion, plastic deformation, and accumulation on both sides, forming furrows through multiple reciprocating frictions. Further, as the plastic deformation continues to develop and accumulate to the coating's limit, the WC and $Cr_7C_3$ hard particles in the coating become loose and fall off, forming pits. Consequently, abrasive wear is the primary wear mechanism for the HVAF coating.

To conduct a more detailed analysis of the changes in coating elements during the friction and wear process, an energy-dispersive X-ray spectroscopy (EDS) analysis was performed on the worn coating, and the results are summarized in Table 5. The wear morphology and an enlarged image of the HVAF coating are presented in Figure 10a,b, respectively. The EDS analysis revealed large amounts of O elements at points 1 and 2, indicating that the HVAF coating underwent oxidation during the friction and wear process. This can be attributed to the high temperature generated by the contact between the WC counterface and the tip of the coating during dry friction, leading to the oxidation of the Cr and W elements in the coating [27]. Moreover, large amounts of C elements were detected at point 3, suggesting that the high hardness of the coating resulted in the transfer of material from the counterface to the coating surface during the friction and wear process. Interestingly, no O elements were detected at point 4, but a high content of C elements was present, indicating that this was likely the result of carbide (WC, $Cr_7C_3$) particles or debris that had been worn away and re-filled into the pits by the friction and compression of the friction pair.

**Table 5.** EDS results of the corresponding region of Stellite 20 coating wear morphology (at%).

| Areas | Elements | | | | | |
|---|---|---|---|---|---|---|
| | Co | Cr | C | O | W | Fe |
| 1 | 9.02 | 9.01 | 31.99 | 44.62 | 2.11 | 0.75 |
| 2 | 8.37 | 7.60 | 29.23 | 45.64 | 7.99 | 0.55 |
| 3 | 17.40 | 16.93 | 51.33 | 6.73 | 2.69 | 0.80 |
| 4 | 18.89 | 20.85 | 48.63 | - | 3.53 | 2.16 |
| 5 | 21.60 | 24.85 | 38.56 | 3.65 | 3.82 | 1.79 |
| 6 | 26.62 | 12.20 | - | 53.45 | 1.75 | 1.25 |
| 7 | 38.97 | 32.72 | - | 14.24 | 4.44 | 2.52 |
| 8 | 40.24 | 32.36 | - | 17.35 | 4.62 | 2.40 |
| 9 | 26.62 | 12.20 | - | 53.45 | 1.5 | 1.25 |
| 10 | 45.14 | 38.96 | - | - | 4.73 | 3.13 |

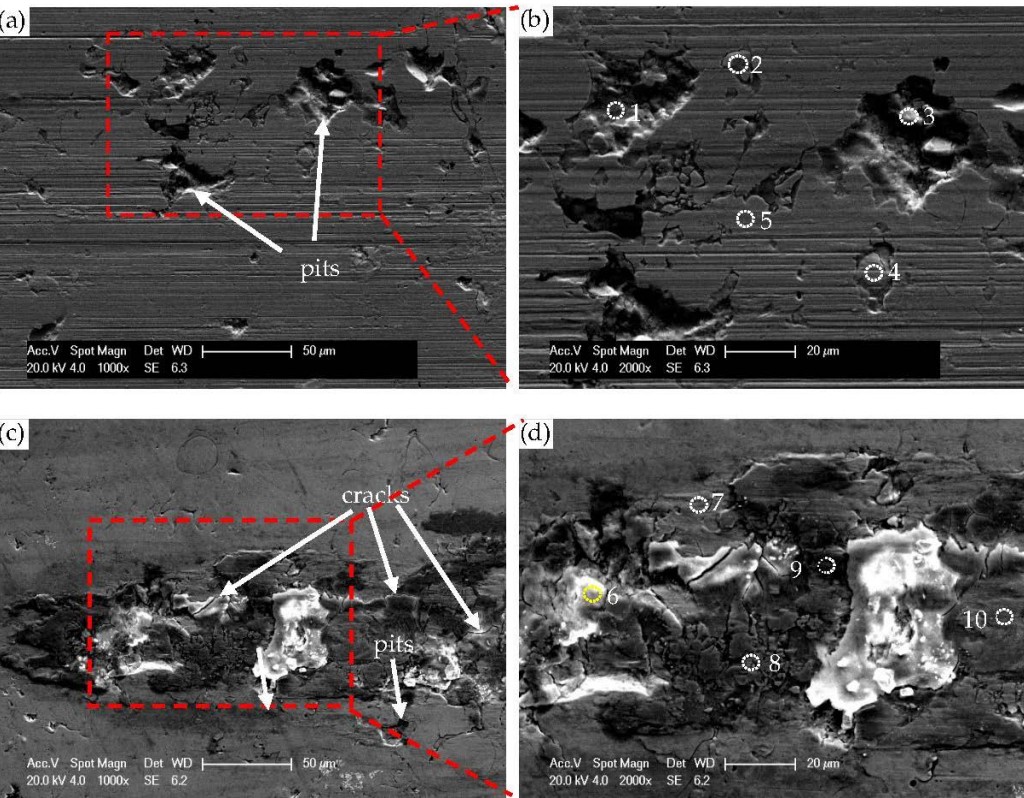

**Figure 10.** The wear morphology of Stellite 20 coating: (**a**,**b**) the wear morphology of HVAF spraying coating; (**c**,**d**) Wear morphology of HVOF sprayed coatings. The numbers in this figure correspond to the 'Areas' in Table 5, indicating that EDS analysis was performed at these locations.

Combining the data presented in Figure 10c,d, and Table 5, it can be inferred that the surface of the HVOF coating contains numerous cracks and Co/Cr oxide debris, as well as traces of coating detachment at the edge of the wear mark. These observations are attributed to the plastic deformation that occurs on the surface of the coating under normal stress. The proliferation and movement of dislocations during plastic deformation results in work hardening and embrittlement in the sub-surface layer, which eventually leads to the origination and propagation of cracks at a certain depth below the coating surface. The resulting thin plate-like wear debris exhibits typical fatigue wear morphology. Additionally, shallow furrows are observed at the edge of the wear mark, indicating that mild abrasive wear accompanies the occurrence of fatigue wear in the HVOF coating.

*3.6. Corrosion Resistance*

The polarization curves of F310H, HVAF, and HVOF coatings were measured in a 3.5% NaCl solution at room temperature, as shown in Figure 11. The corresponding corrosion potential ($E_{corr}$), corrosion current density ($I_{corr}$), and polarization resistance ($R_p$) were extracted and documented in Table 6. The results indicated that both HVAF and HVOF coatings exhibit a pronounced passivation region, indicating the remarkable resistance of the Stellite 20 coating to the harmful effects of corrosive media on the substrate. The corrosion potential of the HVAF coating ($-0.376$ V) is substantially higher than that of the HVOF coating ($-0.532$ V), with an increase of 0.156 V. The corrosion current density of the HVAF coating is one order of magnitude lower than that of the HVOF coating, while the polarization resistance is 2.39 times higher than that of the HVOF coating, affirming the outstanding corrosion resistance performance of the HVAF coating.

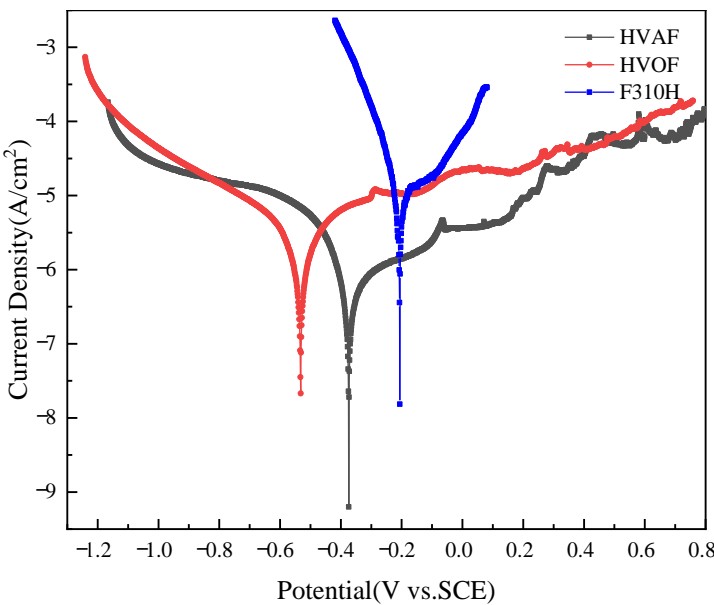

**Figure 11.** Potentiodynamic polarization curves of HVAF and HVOF coatings in comparison with the F310H substrate in 3.5 wt.% NaCl solution.

**Table 6.** Parameter values of potentiodynamic polarization curves of HVAF and HVOF spray coatings in 3.5 wt.% NaCl solution at room temperature.

| Coating | $E_{corr}$/V | $I_{corr}$/A·cm$^{-2}$ | $\beta_a$/mV·dec$^{-1}$ | $-\beta_c$/mV·dec$^{-1}$ | $R_p$/kΩ·cm$^2$ |
|---|---|---|---|---|---|
| F310H | −0.210 | $4.220 \times 10^{-6}$ | 77.00 | 35.90 | $2.520 \times 10^3$ |
| HVAF | −0.376 | $6.268 \times 10^{-7}$ | 493.50 | 114.85 | $6.454 \times 10^4$ |
| HVOF | −0.532 | $1.692 \times 10^{-6}$ | 216.07 | 205.63 | $2.704 \times 10^4$ |

Based on the cross-sectional morphology and XRD results of the HVAF and HVOF coatings presented in Figures 4 and 5, the underlying reason for the discrepancy in corrosion resistance between the coatings can be elucidated. Firstly, it should be noted that the HVOF coatings with oxygen as the combustion-supporting gas have a higher content of $CoCr_2O_4$ metal oxide when compared to HVAF coatings. These metal oxides may impede the development of a dense passive film on the surface of the Stellite 20 coating and may even offer pathways for internal corrosion by electrolytes. Additionally, the HVAF coating has lower porosity, which effectively blocks the diffusion channels of the electrolyte. Consequently, the HVAF coating exhibits superior corrosion resistance performance compared to the HVOF coating.

### 4. Conclusions

1.  The HVOF and HVAF-prepared Stellite 20 coatings exhibit typical thermal spray coating structures, with the HVOF coating composed of partially melted and fully melted zones and a small number of fragmented particles, while the HVAF coating is composed of fragmented particles and partially melted zones, with a small amount of fully melted zones.
2.  Both HVOF and HVAF-prepared coatings show grain refinement compared to the powder raw materials and consist of γ-Co solid solution, ε-Co solid solution, Cr-rich solid solution, $Cr_7C_3$, WC, and trace amounts of $CoCr_2O_4$.
3.  The microhardness of the Stellite 20 coatings prepared by HVOF and HVAF processes is three times higher than that of the F310H substrate. The HVOF coating exhibits a relatively smooth surface and a porosity of 0.41%, and the HVAF coating exhibits a relatively rough surface and a porosity of 0.32%. The HVOF coating has a smaller hardness dispersion, indicating higher mechanical stability.

4.  The wear mechanisms of the coatings are different, with the HVAF coating exhibiting abrasive wear, while the HVOF coating exhibits mainly fatigue wear with slight abrasive wear. The HVAF coating shows better wear resistance due to its higher hardness.

5.  The HVAF coating exhibits lower corrosion current density, measuring $6.268 \times 10^{-7}$ Acm$^{-2}$, one order of magnitude lower than that of the HVOF coating. This is attributed to the lower oxide content and porosity of the HVAF-prepared Stellite 20 coating, resulting in better corrosion resistance compared to the HVOF coating.

**Author Contributions:** Writing—original draft, Z.Z. (Zhiqiang Zhou); formal analysis, Q.C.; investigation, J.Y.; data curation, H.J.; writing—review and editing, D.S.; visualization, Z.Z. (Zhongyun Zhou); project administration, J.H.; funding acquisition, J.H.; supervision, J.H. All authors have read and agreed to the published version of the manuscript.

**Funding:** This research was funded by Major National Science and Technology Projects, grant umber 2019ZX06002026-005, China. The APC was funded by Chongqing Chuanyi Control Valve Co., Ltd., Chongqing, China.

**Institutional Review Board Statement:** Not applicable.

**Informed Consent Statement:** Not applicable.

**Data Availability Statement:** Data sharing is not applicable to this article.

**Conflicts of Interest:** The authors declare no conflict of interest.

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
