# Peer review of "Tribological Properties and Corrosion Resistance of Stellite 20 Alloy Coating Prepared by HVOF and HVAF"

_coatings, doi:10.3390/coatings13040806_

Round 1

Reviewer 1 Report

The main comments that I find useful for improving the quality of the article are presented below:

 - The novelty of the work isn't necessarily clear in the introduction section. Please give the novelty of this manuscript clearly.

 -The authors can control Table 1 and Table 2 for Fe value? and Co value? And figure 5. Please use English alphabet (black xrd).

 - Between the 183 and 192 lines, the authors give the melting points of HVOF and HVAF, but it could not be seen the coating thickness of HVOF and HVAF technique. Please verify the coating thickness.

Author Response

Point 1: The novelty of the work isn't necessarily clear in the introduction section. Please give the novelty of this manuscript clearly. 

Response 1: Thank you for mentioning this important point. Firstly, Compared with the original alloy, the composition of Stellite alloy coatings prepared by plasma welding or laser cladding process will change due to dilution and dissolution of the matrix material, resulting in a decrease in the phase transformation ability and wear resistance of the coating. The Stellite 20 coating prepared using HVOF and HVAF technology in this paper maintains its excellent wear and corrosion resistance due to the low heat input during the preparation process. Secondly, there is currently limited research on the preparation of Stellite 20 coatings using supersonic flame spraying technology. Studying the wear resistance and corrosion resistance of Stellite 20 coatings prepared by HVAF and HVOF processes on F310 heat-resistant stainless steel will provide a theoretical basis for the industrial applications under high temperature and corrosion conditions.

 Point 2: The authors can control Table 1 and Table 2 for Fe value? and Co value? And figure 5. Please use English alphabet (black xrd).

 Response 2: Sorry for our inattention. I have made changes in the paper. See the red section in Table 1, Table 2 and Figure 5.

 Point 3: Between the 183 and 192 lines, the authors give the melting points of HVOF and HVAF, but it could not be seen the coating thickness of HVOF and HVAF technique. Please verify the coating thickness.

Response 3:Thank you for your suggestion. The thickness of the coatings prepared by HVAF and HVOF is between 0.25-0.30 mm, and it is described in lines 225-226 of the article. See changes in red.

Reviewer 2 Report

1. Original submission

1.1. Recommendation

Minor revision

2. Comments to the author

Ms. Ref. No.: Coatings-2330593

Title: Tribological properties and corrosion resistance of Stellite 20 Alloy Coating prepared by HVOF and HVAF

Overview and general recommendation:

This study describes the tribological and corrosion properties of Stellite 20 alloy coatings on F310 heat-resistant stainless steel that were prepared using HVOF and HVAF supersonic flame spraying techniques. To investigate the coatings' microstructure, phase, microhardness, wear, and corrosion resistance, a range of characterization techniques including SEM, EDS, XRD, microhardness, and friction wear-testers were employed. The results indicate that both HVOF and HVAF-prepared coatings exhibit a dense structure with porosity of 0.41% and 0.32%, respectively. The discussion in this study is innovative and novel. Sentences need more clarity and better construction. This manuscript needs minor revisions before can be accepted for publication in the journal. Therefore, the authors are advised to address the following comments carefully.   

The introduction is lack of sufficient background information, which is unable to give the reader detailed background knowledge and possible wide application of this study. The introduction needs to be more emphasized on the research work with a detailed explanation of the whole process considering past, present and future scope. How the present study gives more accurate results than previous studies? It needs to be strengthened regarding recent research in this area with possible research gaps. Research gaps should be highlighted more clearly, and future applications of this study should be added. Many questions need to be addressed, please see the comments.

2.1. Major comments:

1.      There are lots of grammatical and spelling errors. Please correct them.

2.      Most of the references cited are old one. It is recommended to add latest literature in the introduction section. For example, (1) “Raza, A.; Ahmad, F.; Badri, T.M.; Raza, M.R.; Malik, K. An Influence of Oxygen Flow Rate and Spray Distance on the Porosity of HVOF Coating and Its Effects on Corrosion—A Review. Materials 2022, 15, 6329.” https://doi.org/10.3390/ma15186329. (2) Raza, A., Ahmad, F., Badri, T.M., Raza, M.R., Malik, K., Ali, S. (2023). Selection of   Materials Based on Thermo-Mechanical Properties of Thermal Barrier Coatings and Their Failures—A Review. In: Emamian, S.S., Awang, M., Razak, J.A., Masset, P.J. (eds) Advances in Material Science and Engineering. Lecture Notes in Mechanical Engineering. Springer, Singapore. https://doi.org/10.1007/978-981-19-3307-3_22

2.2. Minor comments:

Page 1: Heading for Introduction Section is missing. Please write the heading.

Page 2, Figure 1: It is suggested not to include figure in the Introduction section. If you wish so, please provide the reference.

Page 3: The last columns of Table 1 and Table 2 are in Chinese. Please correct.

Page 3, line 108: “The AISI F310 high-temperature stainless steel substrate material was first purified”. What is the meaning of the word “purified”?

Page 3, Table 3: In the last two columns of Table 3, the correct word is “powder” not “power”. Also, correct the spellings of “Nitrogen” and “distance”.

Page 5, Figure 3: In this figure, the word “picture” is used in the caption. Please replace this word with “SEM image” or another suitable word.

Page 8, line 280: “2.5. Microhardness and Weibull Distribution.” Please correct the section numbering.

Page 12, Figure 10: This figure's red border of dotted lines is out of the figure. Please redraw.

Author Response

Overview and general recommendation:

This study describes the tribological and corrosion properties of Stellite 20 alloy coatings on F310 heat-resistant stainless steel that were prepared using HVOF and HVAF supersonic flame spraying techniques. To investigate the coatings' microstructure, phase, microhardness, wear, and corrosion resistance, a range of characterization techniques including SEM, EDS, XRD, microhardness, and friction wear-testers were employed. The results indicate that both HVOF and HVAF-prepared coatings exhibit a dense structure with porosity of 0.41% and 0.32%, respectively. The discussion in this study is innovative and novel. Sentences need more clarity and better construction. This manuscript needs minor revisions before can be accepted for publication in the journal. Therefore, the authors are advised to address the following comments carefully.   

The introduction is lack of sufficient background information, which is unable to give the reader detailed background knowledge and possible wide application of this study. The introduction needs to be more emphasized on the research work with a detailed explanation of the whole process considering past, present and future scope. How the present study gives more accurate results than previous studies? It needs to be strengthened regarding recent research in this area with possible research gaps. Research gaps should be highlighted more clearly, and future applications of this study should be added. Many questions need to be addressed, please see the comments.

 Response:Thank you very much for taking your time to provide such detailed and valuable comments.  The background information on the research and potential application areas in the introduction section has been improved, and the differences between past and current research have been clearly highlighted. We have made revisions accordingly in the introduction section. See changes in red.

2.1. Major comments:

Point 1.There are lots of grammatical and spelling errors. Please correct them.

Response 1: Thanks for your kind reminders. The grammatical and spelling errors have been rechecked and revised, as detailed in the red part of the paper.

 Point 2. Most of the references cited are old one. It is recommended to add latest literature in the introduction section. 

Response 2: Thank you for your suggestion. We have added latest literature in the introduction section. See changes in red.

2.2. Minor comments:

Point 1: Page 1: Heading for Introduction Section is missing. Please write the heading.

Response 1: Sorry for our inattention. The heading has been added, see line 30 of the article.

Point 2: Page 2, Figure 1: It is suggested not to include figure in the Introduction section. If you wish so, please provide the reference.

Response 2: We would like to thank you for having raised this important point. We believe that keeping this figure in introduction section is conducive to illustrating the principle of supersonic spraying. And this picture was drawn by ourselves.

 Point 3: Page 3: The last columns of Table 1 and Table 2 are in Chinese. Please correct.

Response 3: Sorry for our inattention. This mistakes have been corrected in the text. See changes in Table 1 and Table 2.

 Point 4: Page 3, line 108: “The AISI F310 high-temperature stainless steel substrate material was first purified”. What is the meaning of the word “purified”?

Response 4: We appreciate your detailed considerations at our manuscript. Purified refers to the process of repeatedly washing the F310 substrate with acetone and ethanol absolute to remove impurities such as oxide or oil residues that may remain on the surface of the sample.

 Point 5: Page 3, Table 3: In the last two columns of Table 3, the correct word is “powder” not “power”. Also, correct the spellings of “Nitrogen” and “distance”

Response 5: Sorry for our inattention. These mistakes have been corrected in the Table 3. See changes in red.

 Point 6: Page 5, Figure 3: In this figure, the word “picture” is used in the caption. Please replace this word with “SEM image” or another suitable word.

Response 6: “picture” has been changed to “SEM image” in the caption of Figure 3. See changes in red.

 Point 7: Page 8, line 280: “2.5. Microhardness and Weibull Distribution.” Please correct the section numbering.

Response 7: The “Microhardness and Weibull Distribution” section numbering and subsequent section numbering have been changed. See changes in red.

 Point 8: Page 12, Figure 10: This figure's red border of dotted lines is out of the figure. Please redraw.

Response 8: Thank you very much for taking note of this issue. Corrections have been made to the figure. See changes in red.

Reviewer 3 Report

 This paper investigates the tribological and corrosion properties of Stellite 20 9 alloy coatings on heat-resistant stainless steel F310, which were prepared using a supersonic flame HVOF and HVAF 10 spraying techniques.

The comparison of HVOF and HVAF is sufficiently described. Fig.3 is confusing and is it necessary? Fig. 3 shows morphology like in Fig 4, but at high magnification.

As written in line 341: The friction coefficient of the HVOF coating exhibits higher stability compared to the HVAF coating. Can the authors check whether it is also related to the structure of the coating?

Very nice and interesting article. Scientific results are new.

Author Response

Point 1: The comparison of HVOF and HVAF is sufficiently described. Fig.3 is confusing and is it necessary? Fig. 3 shows morphology like in Fig 4, but at high magnification.

Response 1: We believe that the presence of Figure 3 is necessary because it displays the rough surface morphology of the coating, while Figure 4 shows the cross-sectional morphology of the coating after grinding and polishing. The coexistence of both figures better illustrates the structural differences between coatings prepared by HVAF and HVOF.

 Point 2: As written in line 341: The friction coefficient of the HVOF coating exhibits higher stability compared to the HVAF coating. Can the authors check whether it is also related to the structure of the coating?

Response 2: Thank you for your suggestion. The current experimental data can only confirm that higher stability of the friction coefficient of the HVOF coating is due to its more uniform overall hardness distribution compared to the HVAF coating. Both coatings have a typical thermal spray layer structure, and further research is needed to determine the differences in structure between the two coatings and their effects on the friction coefficient.

 Point 3: Very nice and interesting article. Scientific results are new.

Response 3: Thanks very much. We appreciate you for your precious time in reviewing our paper and providing valuable comments.